# Mucosal immunity of mannose-modified chitosan microspheres loaded with the nontyepable *Haemophilus influenzae* outer membrane protein P6 in BALB/c mice

Yushuai Ma[1]☺, Ying Zhao[1]☺, Rui Chen[1], Wanru Sun[1], Yanxia Zhang[2], Haixia Qiao[2], Yueli Chang[2], Shaoping Kang[3], Yutuo Zhang[2]*

**1** Institute of Pathogen Biology and Immunology, Hebei North University, Zhangjiakou, Hebei Province, China, **2** Department of Microbiology, Hebei North University, Zhangjiakou, Hebei Province, China, **3** Department of Immunology, Hebei North University, Zhangjiakou, Hebei Province, China

☺ These authors contributed equally to this work.
\* yvtuozhang@163.com

**Data Availability Statement:** All relevant data are within the manuscript and its Supporting information files.

## Abstract

Nontypeable *Haemophilus influenzae* (NTHi) is a common opportunistic pathogen that colonizes the nasopharynx. NTHi infections result in enormous global morbidity in two clinical settings: otitis media in children and acute exacerbation of chronic obstructive pulmonary disease (COPD) in adults. Thus, there is an urgent need to design and develop effective vaccines to prevent morbidity and reduce antibiotic use. The NTHi outer membrane protein P6, a potential vaccine candidate, is highly conserved and effectively induces protective immunity. Here, to enhance mucosal immune responses, P6-loaded mannose-modified chitosan (MC) microspheres (P6-MCMs) were developed for mucosal delivery. MC (18.75%) was synthesized by the reductive amination reaction method using sodium cyanoborohydride ($NaBH_3CN$), and P6-MCMs with an average size of 590.4±16.2 nm were successfully prepared via the tripolyphosphate (TPP) ionotropic gelation process. After intranasal immunization with P6-MCMs, evaluation of humoral immune responses indicated that P6-MCMs enhance both systemic and mucosal immune responses. Evaluation of cellular immune responses indicated that P6-MCMs enhance cellular immunity and trigger a mixed Th1/Th2-type immune response. Importantly, P6-MCMs also trigger a Th17-type immune response. They are effective in promoting lymphocyte proliferation and differentiation without toxicity in vitro. The results also demonstrate that P6-MCMs can effectively induce MHC class I- and II-restricted cross-presentation, promoting CD4+-mediated Th immune responses and CD8+-mediated cytotoxic T lymphocyte (CTL) immune responses. Evaluation of protective immunity indicated that immunization with P6-MCMs can reduce inflammation in the nasal mucosa and the lung and prevent NTHi infection. In conclusion, MCMs are a promising adjuvant-delivery system for vaccines against NTHi.

**Funding:** This work was supported by Hebei Province University Science and Technology Research Key Project (No. ZD2016030); Hebei Province Education Department Youth Fund Project (No. QN2015024).

**Competing interests:** The authors have declared that no competing interests exist.

## Introduction

Gram-negative *Haemophilus influenzae* (Hi), a common opportunistic pathogen in the clinic, colonizes the nasopharynx in humans under normal conditions. This species can cause acute suppurative infection in individuals with low resistance or an unbalanced local microecological environment, and it can even cause secondary infections such as meningitis, pneumonia, sepsis, sinusitis, otitis media, and recurrent respiratory infections [1]. Hi can be divided into encapsulated strains and nonencapsulated strains; the latter are designated nontypeable (NTHi) and have been listed as one of 12 high-priority bacterial pathogens by the World Health Organization [2]. Six serotypes (a, b, c, d, e, and f) of encapsulated *H. influenzae* have been identified, and type b (Hib) is responsible for most invasive disease as a major virulent pathogen [3].

There is an urgent need to design and develop effective vaccines for NTHi due to the lack of effective vaccines as well as the spread and prevalence of NTHi worldwide. The vaccine effectiveness reportedly is variable in some infants, some age groups, or some forms (infection vs. invasive disease). While the initial impact of the vaccine was impressive when it was released, there has been an increase in Hib infections over the years despite vaccination control programs [4]. However, these conjugate vaccines have no immune effect on infections caused by NTHi without a polysaccharide capsule. As a result, NTHi has become the major pathogen causing invasive Hi infection, which has attracted more attention from researchers because of the increased prevalence and severity of infections caused by NTHi [5, 6], such as otitis media (OM) in children, cystic fibrosis (Cf), community-acquired pneumonia in children, chronic bronchitis, conjunctivitis, acute exacerbation of chronic obstructive pulmonary disease (COPD) in adults, and urinary tract infections [7, 8]. In further research on NTHi, several outer membrane proteins have been identified as potential vaccine candidates, of which the outer membrane protein P6 that is expressed in all Hi strains is highly conserved and can induce protective immunity [9–13].

The respiratory mucosa is the first barrier to prevent Hi invasion, and it provides host defense at mucosal surfaces based on the mucosa-related immunoglobulin, IgA. Intranasal immunization can induce mucosal immune responses in nasal-associated lymphoid tissue (NALT), stimulating distal IgA-mediated mucosal immune responses (gastrointestinal, respiratory, and urogenital) and triggering both systemic humoral and cellular immune responses. Therefore, intranasal immunization is probably the most effective immune route for P6 because of the nasopharyngeal colonization of Hi [14–17].

Although mucosal delivery is a well-documented and highly effective route for the stimulation of local and systemic immunity, soluble P6 induces a weak immune response when administered by mucosal routes, which require a mucosal adjuvant or a delivery system. Recently, mannosylated chitosan microspheres have received attention as an adjuvant-delivery system to enhance the mucosal immune response to specific antigens [18–20]. Chitosan, a biocompatible and degradable polysaccharide, can be degraded and absorbed completely by the body. Chitosan microspheres carrying antigens reduce self-clearing of soluble antigens from the nasal mucosa via their adhesion and high permeability and provide sustained immune activity via controlled release of immunogen [21, 22]. Importantly, chitosan microspheres can be efficiently phagocytized by M cells and taken up by dendritic cells (DCs) and macrophages (MΦs), inducing mucosal and systemic specific responses without toxic side effects [21]. Mannose receptors (MRs) have been used in the delivery systems of various vaccines and are present on antigen-presenting cells (APCs), such as DCs and MΦs [23, 24]. Mannose is currently the only glycotrophic nutrient in the clinic, and mannosylated carriers can be captured through receptor-mediated endocytosis by targeting MR, which improves the antigen uptake

efficiency of APC and is involved in MHC class I- and II-restricted antigen presentation, bolstering cellular immune responses [25–27].

In this study, mannosylated chitosan was obtained via NaBH$_3$CN catalysis, and chitosan microspheres loaded with P6 (P6-CMs) and mannose-modified chitosan microspheres loaded with P6 (P6-MCMs) were prepared by ion condensation. Vaccination and immune protection experiments were performed via intranasal administration in BALB/c mice. The changes in humoral immunity and cellular immunity were measured to evaluate the immune effect of the microsphere vaccines.

## Materials and methods

### Materials

NTHi (ATCC49247) was purchased from Beijing institute of BeNa Biotechnology (Beijing, China), PGEX-6p2 and *E. coli* XL1-Blue were obtained from Hebei Medical University (Shijiazhuang, China), and chitosan (low molecular weight, deacetylation 75–85%), mannose, sodium cyanoborohydride (NaBH$_3$CN), sodium tripolyphosphate (TPP), and D-glucosamine hydrochloride were purchased from Sigma-Aldrich (St. Louis, MO, USA). Six-week-old female specific pathogen-free (SPF) BALB/c mice were cared for in the Laboratory Animal Center of Life Science Research Center (Hebei North University, Zhangjiakou, China), and the mice were used in accordance with the policies and regulations related to the care and use of laboratory animals.

### Preparation of vaccine

**Preparation of loaded antigens.**   The P6 gene was amplified from NTHi (ATCC49247) template DNA by PCR and inserted into the prokaryotic vector PGEX-6p2 to construct the recombination plasmid PGEX-6p2/P6, which was transformed into the expression host strain *E. coli* XL1-Blue. Then, IPTG was used to induce the expression of the protein [28]. The loaded antigen, P6, was obtained by purification of glutathione S-transferase (GST)-P6 using GSTrap 4B (GE Healthcare Bio-Sciences AB, Sweden) and removal of the GST-tag using PreScission Protease (GE Healthcare Bio-Sciences AB, Sweden). SDS-PAGE and Western blotting were used to verify P6.

**Mannose-modified chitosan (MC) synthesis.**   MC was synthesized by the reductive amination reaction method (Fig 1), as previously reported [29]. Chitosan (C) was dissolved fully in 1% aqueous acetic acid (pH = 5.5), and a solution of mannose and NaBH$_3$CN was added to the viscous solution above. The reaction proceeded with gentle stirring at room temperature for 48 h followed by dialysis for 5 days and lyophilization. The content of free amino acids in C and MC (C $_{Free\ amino}$ and MC $_{Free\ amino}$) was measured with ninhydrin (Sigma-Aldrich, USA), a reagent normally used to quantify free amino acids. The method was as follows: solutions of

**Fig 1. Synthetic route of mannose-modified chitosan derivatives.**

chitosan and MC (0.1 mg ml$^{-1}$) were fully dissolved in 3% aqueous acetic acid and then mixed with 1 ml of sodium acetate (2 M, pH 5.5) and 1 ml of 1% ninhydrin in a tube in boiling water for 20 min. Then, the absorbance at 570 nm ($A_{570 \, nm}$) was read in a 722 G spectrophotometer (INESA, Shanghai, China), and The content of free amino acids was quantified according to a standard curve generated with D-glucosamine hydrochloride (100% free amino) [30]. The degree of substitution (DS) of MC was calculated by the formula DS = ($C_{Free \, amino}$-$MC_{Free \, amino}$)/$C_{Free \, amino}$.

**Preparation of the vaccine: Chitosan microspheres loaded with P6 (P6-CMs) and MC microspheres loaded with P6 (P6-MCMs).** Microspheres were prepared by the ionotropic gelation process following the report of Jiang et al. [31]. Briefly, chitosan and MC were dissolved in 1% (v/v) acetic acid solution. The pH of chitosan was adjusted to 5.4, and the concentration was adjusted to 0.2% (w/v). The TPP solution (1 mg/ml) was dropped into the chitosan solution according to the appropriate ratio (chitosan:TPP = 6:1) with magnetic stirrers for 40 min. While equivalent MC microspheres were formed, the TPP solution (1 mg/ml) was dropped into 0.25% (w/v) MC solution (pH 5.6) according to the appropriate quantity ratio (chitosan:TPP = 4:1) with magnetic stirrers for 40 min. The formed microspheres were washed with deionized water by centrifugation at 14000 rpm for 20 min. P6 was loaded on the microspheres in PBS (pH 7.4) and incubated for 12 h at 25˚C with continuous shaking. Then, the loading capacity (LC) was quantified by the BCA protein assay method.

**Characterization of P6-CMs and P6-MCMs.** The particle size and zeta of the microsphere vaccines were measured using a Zetasizer dynamic light scattering instrument (Nano-ZS90, Malvern Instruments Ltd., UK), and the surface morphology was observed using a scanning electron microscope (S-3400N, Hitachi, Japan) after being gold coated. The in vitro release study was performed at 37˚C in PBS (pH 7.4) with shaking to determine the release rate of P6.

## Vaccination of mice

The mice were randomly divided into five groups, PBS, MCMs, P6, P6-CMs, and P6-MCMs, and intranasal immunization of the mice was performed on days 0, 14, and 28 by dropping 20 μl of PBS containing 20 μg of P6 antigen according to the experimental group: P6, P6-CMs, and P6-MCMs via intranasal drip. The mice in the MCM group were immunized with 20 μl of PBS containing bed volumes of MCMs equal to those in the P6-MCM group. Specimens were collected, and several immune indexes were detected in the second week after the last vaccination. All experiments were approved by the Animal Utilization Committee of Hebei North University and were in accordance with EU Directive 2010/63/EU for animal experiments. Mice were anesthetized via intraperitoneal injection of sodium pentobarbital (1%, 50 mg/kg) and sacrificed via rapid dislocation of the necks. All efforts were made to minimize animal suffering and to reduce the number of animals used.

**Humoral immune responses.** P6-specific IgA and IgG were measured by ELISA as a reflection of systemic and mucosal immunity. Briefly, diluted samples of serum, nasal cavity lavage fluid and lung lavage fluid were added to 96-well plates coated with P6 as the primary antibody, and the reaction was developed with the substrate TMB after incubation with HRP-conjugated goat anti-mouse IgA/IgG (Biosharp, Beijing, China) and quenched with 2 M $H_2SO_4$. Finally, the optical density at 450 nm ($OD_{450}$) was read with a spectrophotometer (Multiskan GO, Thermo, Finland). Blood samples were collected via the retro-orbital sinuses into drop tubes. The collection methods of lavage are as follows: The mice were sacrificed via rapid dislocation of the necks and dissected subsequently, the trachea was exposed and ligated from the middle section. After then, 500 μl PBS was injected in the

trachea toward the lungs, thus the lungs lavage were obtained from the trachea after 5 minutes with gently kneading lung. Injecting 500 μl PBS toward the nasopharynx the same way, collecting the fluid flowing out of the nasal cavity, thus the nasal cavity lavage were obtained after repeating three times.

**Cellular immune responses.** Measurement of cytokines in spleen lymphocytes.

Spleen tissue were homogenized with RPMI-1640, and lymphocytes were isolated from spleen tissue with lymphocyte separation medium, which were adjusted to a concentration of $2 \times 10^6$ cells/ml and cultured with RPMI-1640 in 96-well plates at 37°C under 5% $CO_2$ for 72 h. In addition, vaccines containing 5 μg/ml P6 were added to stimulate the production of IFN-γ, IL-2, IL-4, IL-5 and IL-17a, and ELISA kits (Multi Science, Hangzhou, China) were used for detection.

**Spleen lymphocyte proliferation assay.** Spleen lymphocytes were obtained from previous methods and isolated from spleen tissue with lymphocyte separation medium. Vaccines containing 5 μg/ml P6 were applied to stimulate the proliferation of lymphocytes ($5 \times 10^6$ cells/ml) plated in 96-well plates at 37°C under 5% $CO_2$ for 56 h. Then, the cells were incubated sequentially with CCK-8 for 4 h. Finally, $A_{450}$ was read with a spectrophotometer. The following formula was used to calculate the stimulation index (SI): SI = $A_{450}$ of stimulating group / $A_{450}$ of control group.

**T lymphocyte subpopulation assay.** Lymphocytes were isolated from spleen tissue homogenate with lymphocyte separation medium. Then, the lymphocytes were adjusted to a concentration of $1 \times 10^7$ cells/ml, and 100 μl of cells were stained with APC-Cy™7 Rat Anti-Mouse CD3, FITC Rat Anti-Mouse CD4 and PE Rat Anti-Mouse CD8a (BD Biosciences, San Diego, US) at room temperature for 30 min. The cells were analyzed with a FACSCalibur flow cytometer (BD Biosciences, San Jose, USA) to identify the CD3$^+$, CD4$^+$, and CD8$^+$ T cell subpopulations.

## Evaluation of protective immune responses

To assess the protective immune effect of the microsphere vaccines in BALB/c mice, the mice were anesthetized and challenged with NTHi (ATCC 49247) in a bacterial suspension containing an dose (LD100) ($1 \times 10^8$ CFU/ml) via intranasal drip after the last immunization. One week later, The mice were sacrificed via rapid dislocation of the necks, nasal mucosa and lung tissue were obtained to prepare pathological sections, histopathologic examination was performed by hematoxylin-eosin staining. Differences between groups were analyzed by pathology scores [32]. To score lung inflammation and damage, the following parameters: edema, interstitial inflamamation, intra-alveolar inflammation, endothelialitis, hyperemia, degree of inflammatory cell infiltration. Each parameter was graded from 0 (absent) to 4 (severe). Nasal mucosa were scored according to the following parameters: presence and degree of inflammation cell infiltration, presence and degree of the cilia of the nasal mucosa disappearing, degree of looseness of the columnar epithelial cells arranging. Each parameter was graded from 0 (absent) to 3 (severe). The total pathology scores were expressed as the sum of the score for all parameters.

## Statistical analysis

All statistical analyses were performed with SPSS 25.0 software, and the levels of antibodies and cytokines were analyzed by ANOVA test. All data are expressed as the mean ± standard deviation (SD). Differences were considered statistically significant when $P<0.05$ (*$P<0.05$, ** $P<0.01$ and *** $P<0.001$).

## Results

### Cloning and expression of the loaded antigen

462 bp P6 DNA fragments were amplified by PCR and identified by 1.0% agarose gel electro-phoresis (Fig 2A); after cloning and expression, the size of the soluble protein P6 was approximately 16 kDa after purification and GST tag removal, and SDS-PAGE and Western blot experiments were performed to verify the molecular weight and specificity of polyclonal antibodies against P6 (Fig 2B and 2C).

### Preparation and evaluation of microsphere vaccines

Chitosan was modified covalently with hydrophilic mannose using $NaBH_3CN$. The degree of substitution (DS) of MC was 18.75%, which was determined according to the quantitative difference in free amino acids between C and MC (Table 1). A standard curve is shown in Fig 3C. A value of 100% free amino was assigned to the slope corresponding to the different volumes of D-glucosamine solution (0.1 mg ml$^{-1}$), giving the content level of free amino in C and MC based on the $A_{570 \, nm}$ of the reaction product of amino groups with ninhydrin (Table 1).

The chitosan and MC microspheres (CMs and MCMs) formed as a result of complex coacervation based on the ionotropic gelation of chitosan with TPP anions. The protein P6 was loaded onto the microspheres to prepare P6-CMs and P6-MCMs, and the total loading capacity was 7.13±0.39 mg P6 per milliliter bed volume CMs or 9.52±0.29 mg P6 per milliliter bed volume MCMs. Scanning electron micrographs present some spherical solid dispersion, and P6-MCMs are larger than P6-CMs (Fig 3A). As measured and analyzed for the size distribution (Fig 3B), the average particle sizes of P6-CMs and P6-MCMs were 463.7±15.1 nm and 590.4±16.2 nm, respectively. Moreover, the other characteristics of the microsphere vaccines are shown in Table 2.

The release rate of P6 from the loaded microspheres in vitro was determined by BCA protein assay. As shown in Fig 2D, the continuous release profiles indicated that the release rate of P6 from P6-CMs increased after modification with hydrophilic mannose.

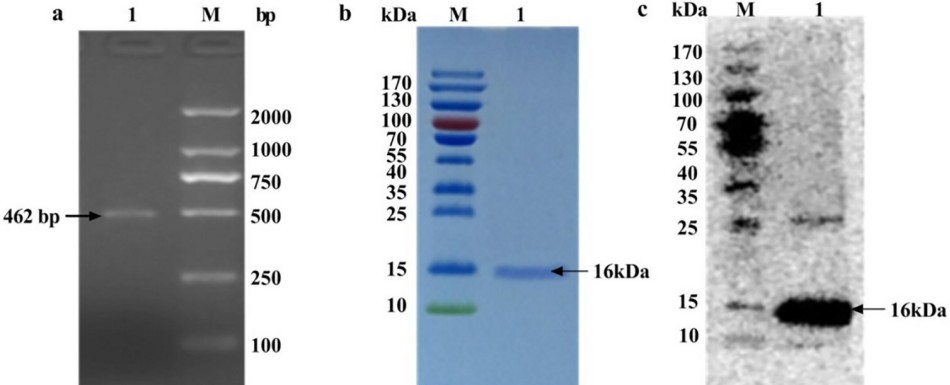

**Fig 2. Gene cloning and expression of the loaded antigen P6. A** Amplification product for the NTHi-P6 gene by PCR. *Lane 1* P6 gene, *Lane M* DNA ladder DL2000. **B** SDS-PAGE gel analysis of tag-removed P6 protein expressed from the NTHi-P6 gene. *Lane 1* P6 protein, *Lane M* prestained protein ladder. **C** Western blot analysis of tag-removed P6. *Lane 1* polyclonal antibodies against P6, *Lane M* prestained protein ladder.

**Table 1. The Degree of Substitution (DS) of Mannose-Modified Chitosan (MC) (mean ± SD, n = 3).**

| Sample | $A_{570\ nm}$ | Content analysis of Free amino | DS (%) |
|---|---|---|---|
| Chitosan | 0.167±0.011 | 0.032±0.002 | |
| MC | 0.137±0.019 | 0.026±0.003 | 18.75 |

## P6-specific systemic and mucosal immune responses

Serum, nasal lavage fluid and lung lavage fluid of vaccinated mice was collected in the second week after the final immunization to detect the levels of P6-specific antibodies in different treatment groups. As expected, specific antibody responses in the P6 group showed a significant increase compared with those in the PBS group ($P<0.01$) (Fig 4). However, as shown in Fig 4A, the level of systemic IgG antibody in the serum of the P6-MCM group was significantly higher than that in the group treated with P6 ($P<0.01$) or MCMs ($P<0.001$), and that in the P6-CM group was also increased significantly compared with that in the P6 group ($P<0.05$). Regarding mucosal immune responses, the mucosal P6-specific IgA antibody in nasal and lung lavage fluid was measured, and the results are shown in Fig 4B. In nasal and lung lavage fluid, the P6-specific antibody levels in the P6-MCM group were significantly higher than those in the P6 and MCM groups ($P<0.001$), and the levels in the P6-CM group were also significantly higher than those in the P6 group ($P<0.01$). These results indicate that the groups administered microspheres loaded with P6 showed enhanced immune responses. Specifically, compared with the P6-CM group, the P6-MCM group showed a significant antibody response in terms of the levels of IgG ($P<0.05$) and IgA ($P<0.01$), suggesting that microsphere vaccines modified with mannose enhance humoral immunity, especially mucosal immunity.

## Measurement of cytokines produced by spleen lymphocytes

The culture supernatants of spleen lymphocytes were obtained to detect the levels of Th1-type (IL-2 and IFN-γ), Th2-type (IL-4 and IL-5) and Th17-type (IL-17) cytokines. A comparison between the P6-MCM group and the P6 or MCM group showed that there were significant differences ($P<0.05$) in the levels of IL-2 (Fig 5A), IFN-γ (Fig 5B), IL-4 (Fig 5C) and IL-5 (Fig 5D), suggesting that microsphere vaccines modified with mannose not only enhance cellular immunity but also trigger a mixed Th1/Th2-type immune response. The P6-CM group presented a significant difference compared with the P6 group only in the level of IL-4 ($P<0.01$) (Fig 5C), which indicates that chitosan microsphere vaccines induce a Th2-type immune response. Moreover, IL-17 levels were increased most significantly in the P6, P6-CM and P6-MCM groups compared with their corresponding control groups, which indicates that they induce the differentiation of Th17 cells and Th17-type immune responses, further demonstrating the development of mucosal immunity and the feasibility of these microspheres as a vaccine.

## Spleen lymphocyte proliferation assay

The stimulation index of spleen lymphocytes was detected to reflect lymphocyte proliferation ability in different treatment groups. As shown in Fig 5F, the stimulation index in the P6 group was significantly higher than that in the PBS group, and the stimulation index of the P6-MCM group increased significantly compared with that of the P6 group but was not significantly higher than that of the P6-CM group. The results indicate that microspheres modified with mannose are effective in promoting lymphocyte proliferation.

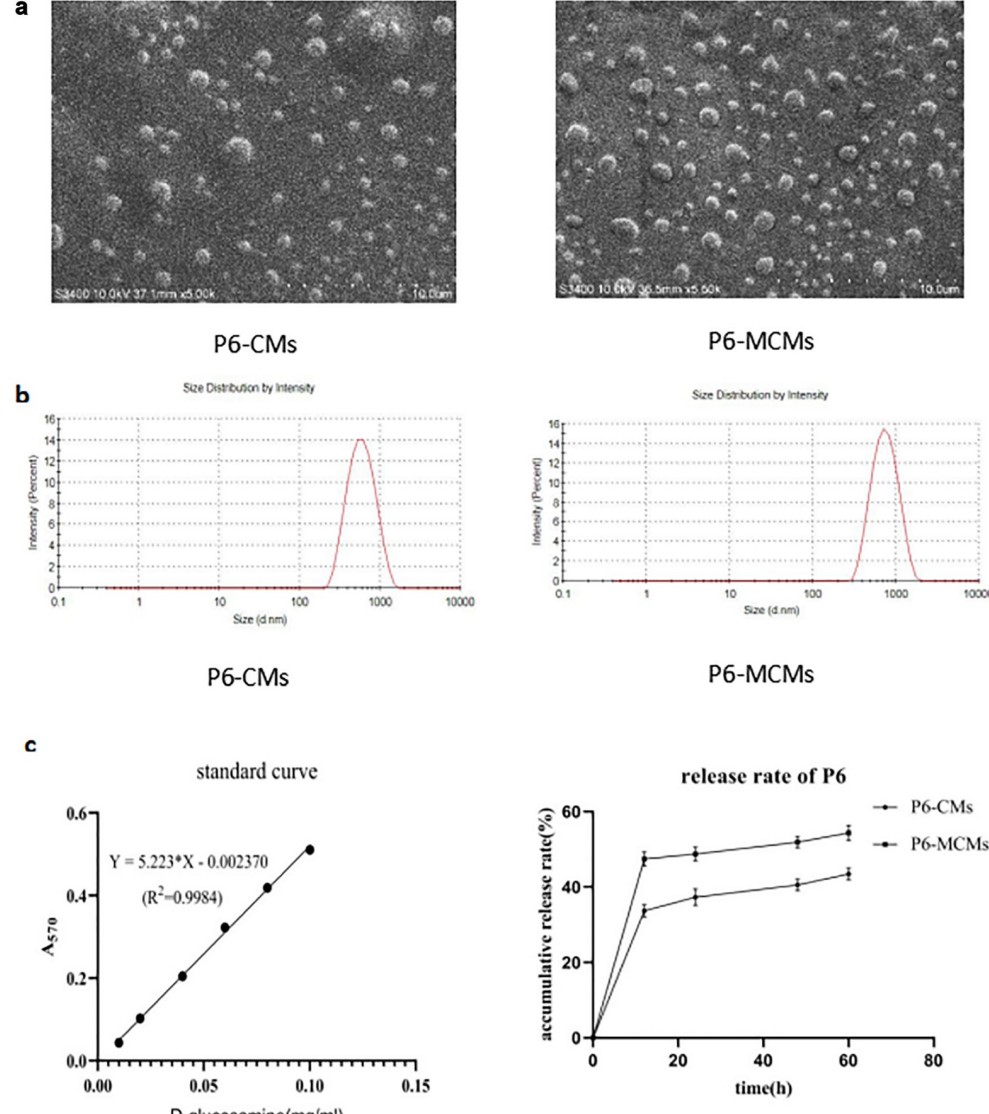

**Fig 3. Characteristics and evaluation of P6-CMs and P6-MCMs. A** SEM images of P6-CMs and P6-MCMs (5000x); the scale bar represents 10 μm. **B** Particle size distribution by intensity (percent) of P6-CMs and P6-MCMs. **C** Standard curve for D-glucosamine hydrochloride (100% free amino, 0.1 mg ml$^{-1}$) at different volumes. The slope was 5.2233, $R^2$ = 0.9984. **D** Continuous release profiles of P6-CMs and P6-MCMs at different times in vitro. Mean ± SD, n = 3.

## T lymphocyte subpopulation assay

The subpopulations of lymphocytes separated from splenocytes were characterized with a FACSCalibur flow cytometer. Fig 6A shows the alterations in CD3$^+$, CD3$^+$CD4$^+$ and

**Table 2. Characteristics of the loaded microspheres (mean ± SD, n = 3).**

| Vaccine | Size-average (nm) ±SD | Size-peak (nm) ±SD | Zeta potential (mV) ±SD | Loading capacity (P6/ml microsphere bed volume) ±SD |
|---|---|---|---|---|
| P6-CMs | 463.7±15.1 | 614.1±15.5 | 10.55±0.64 | 7.13±0.39 |
| P6-MCMs | 590.4±16.2 | 783.0±16.1 | 8.03±0.72 | 9.52±0.29 |

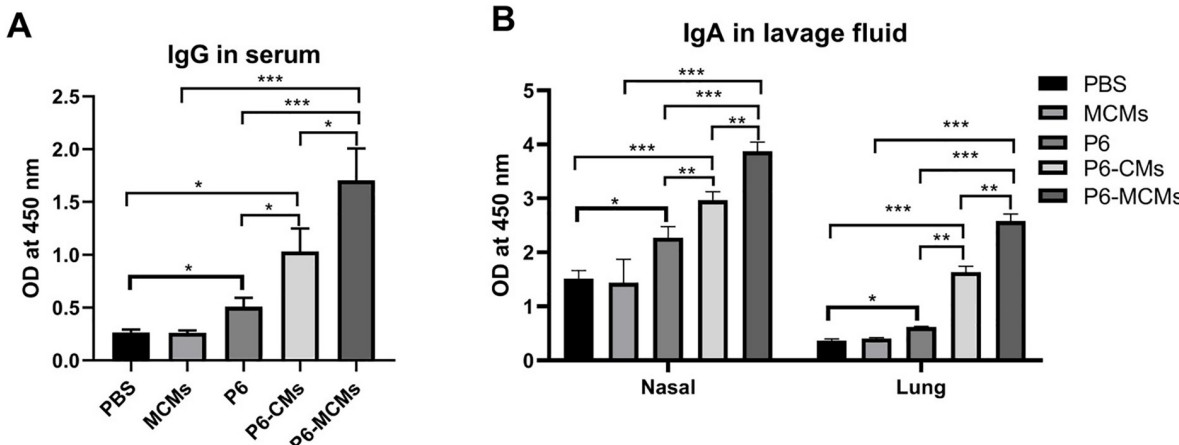

**Fig 4. Analysis of P6-Specific antibody levels in different treatment groups of immunized mice. A** The levels of P6-specific systemic IgG antibody in serum. **B** The levels of the P6-specific mucosal IgA antibody in nasal and lung lavage fluid. The antibody levels were indirectly presented in the form of optical density (OD) values. Values are the mean ± SD, n = 3. Significant differences were expressed as *$P<0.05$, **$P<0.01$, ***$P<0.001$.

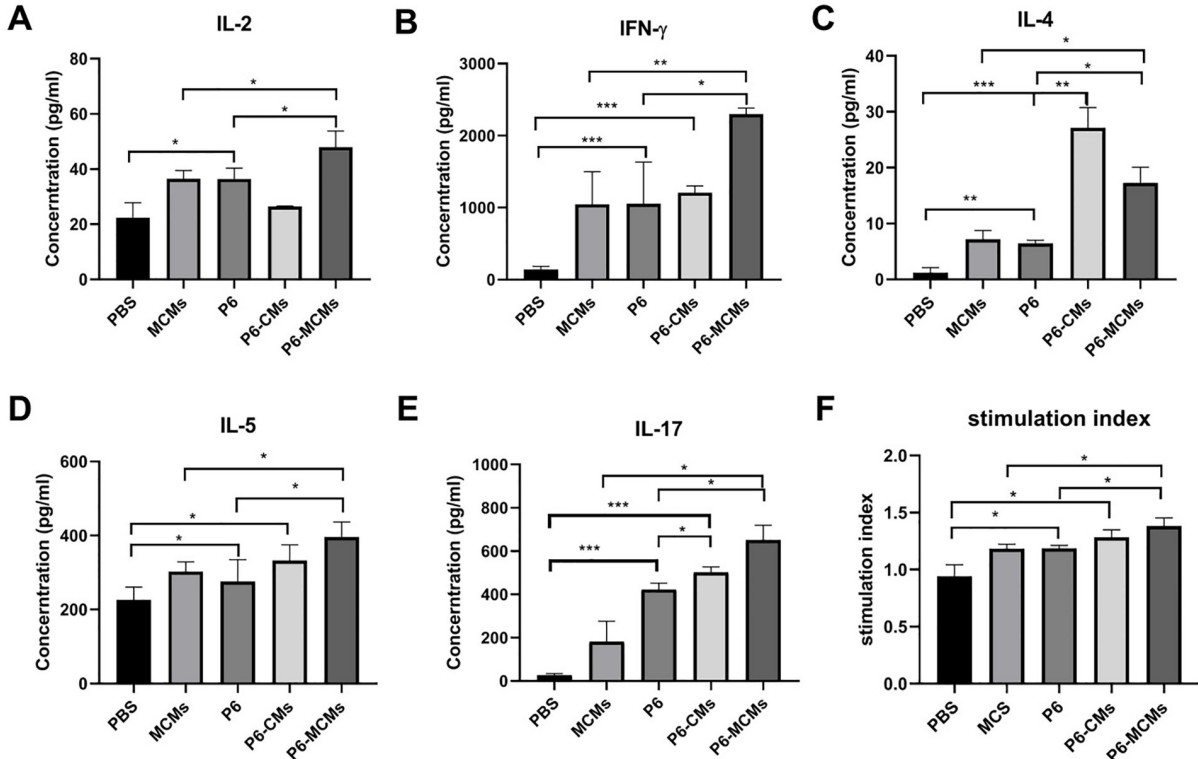

**Fig 5. Analysis of cytokine levels in spleen lymphocyte culture supernatants and T lymphocyte proliferation assays.** The levels of IL-2 (**A**), IFN-γ (**B**), IL-4 (**C**), IL-5 (**D**), and IL-17 (**E**) in lymphocyte culture supernatants. (**F**) Stimulation index of spleen lymphocytes determined according to the absorbance at 450 nm of the stimulated group and the control group. Values are the mean ± SD, n = 3. Significant differences were expressed as *$P<0.05$, **$P<0.01$, ***$P<0.001$.

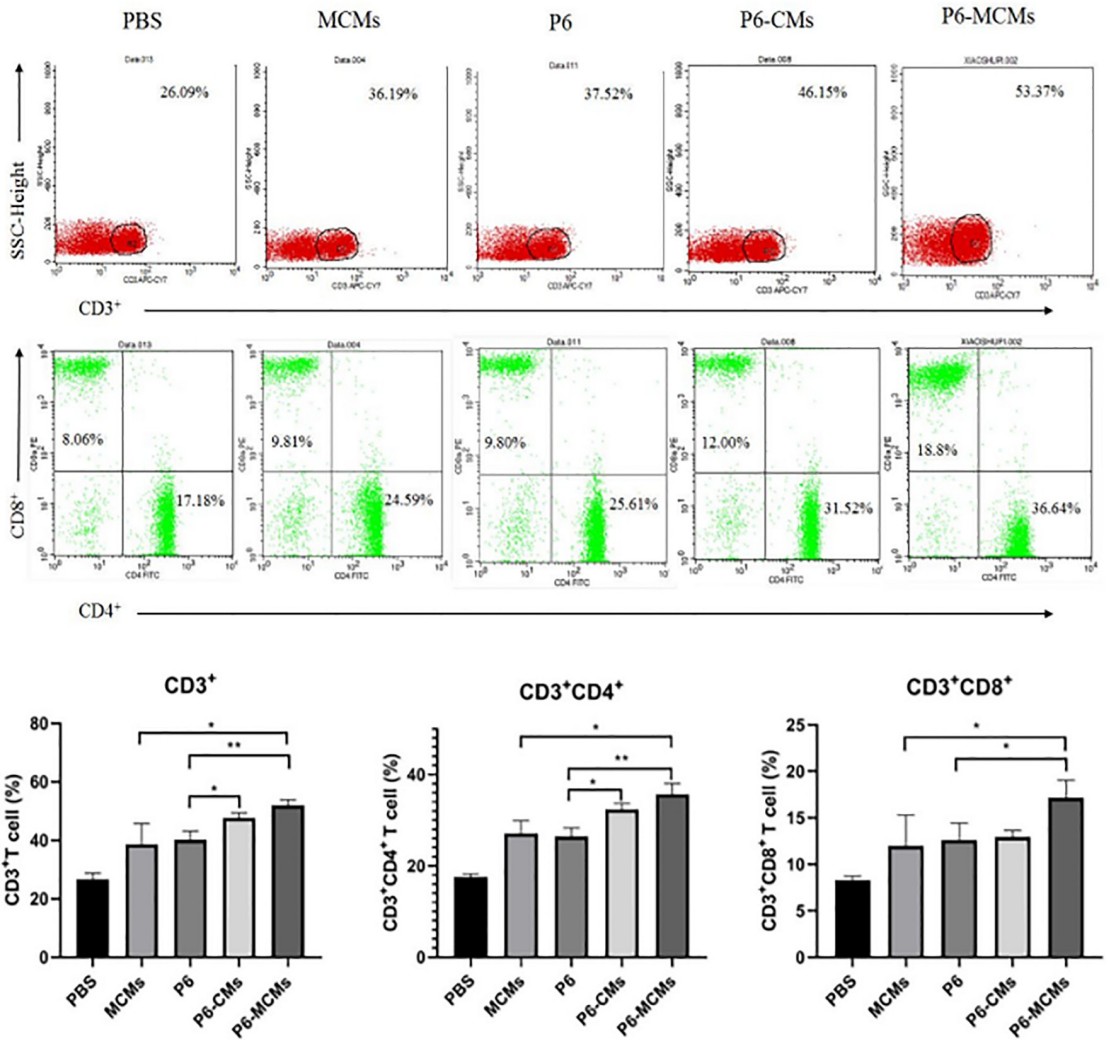

**Fig 6. Flow cytometric analysis of spleen T lymphocyte subsets in different groups of immunized mice. A** "Three-Color, Dual Anchor" gating strategy to identify the lymphocyte subsets (CD3+, CD4+ and CD8+); the proportions are shown. Values are expressed as a percentage. Cells were stained with APC-Cy™7 Rat Anti-Mouse CD3, FITC Rat Anti-Mouse CD4 and PE Rat Anti-Mouse CD8a. **B** Comparison of CD3+ T cell proportions in spleen lymphocytes. **C** Comparison of CD3+CD4+ T cell proportions in spleen lymphocytes. **D** Comparison of CD3+CD8+ T cell proportions in spleen lymphocytes. Mean ± SD, n = 3. $^*P<0.05$, $^{**}P<0.01$.

CD3+CD8+ cell proportions in different groups after intranasal immunization. A significant increase in CD3+ T cell levels was observed in the P6-CM ($P<0.05$) and P6-MCM ($P<0.01$) groups (Fig 6B). In addition, the proportions of CD3+CD4+ ($P<0.01$) and CD3+CD8+ ($P<0.05$) T cells were increased in the P6-MCM group, but only the CD3+CD4+ T cell proportions were increased in the P6-CM group (Fig 6C and 6D). In other words, these results demonstrate that mannose-modified chitosan microspheres can effectively induce MHC class I- and II-restricted antigen presentation, resulting in CD4+-mediated Th immune responses and CD8+-mediated cytotoxic T lymphocyte (CTL) immune responses.

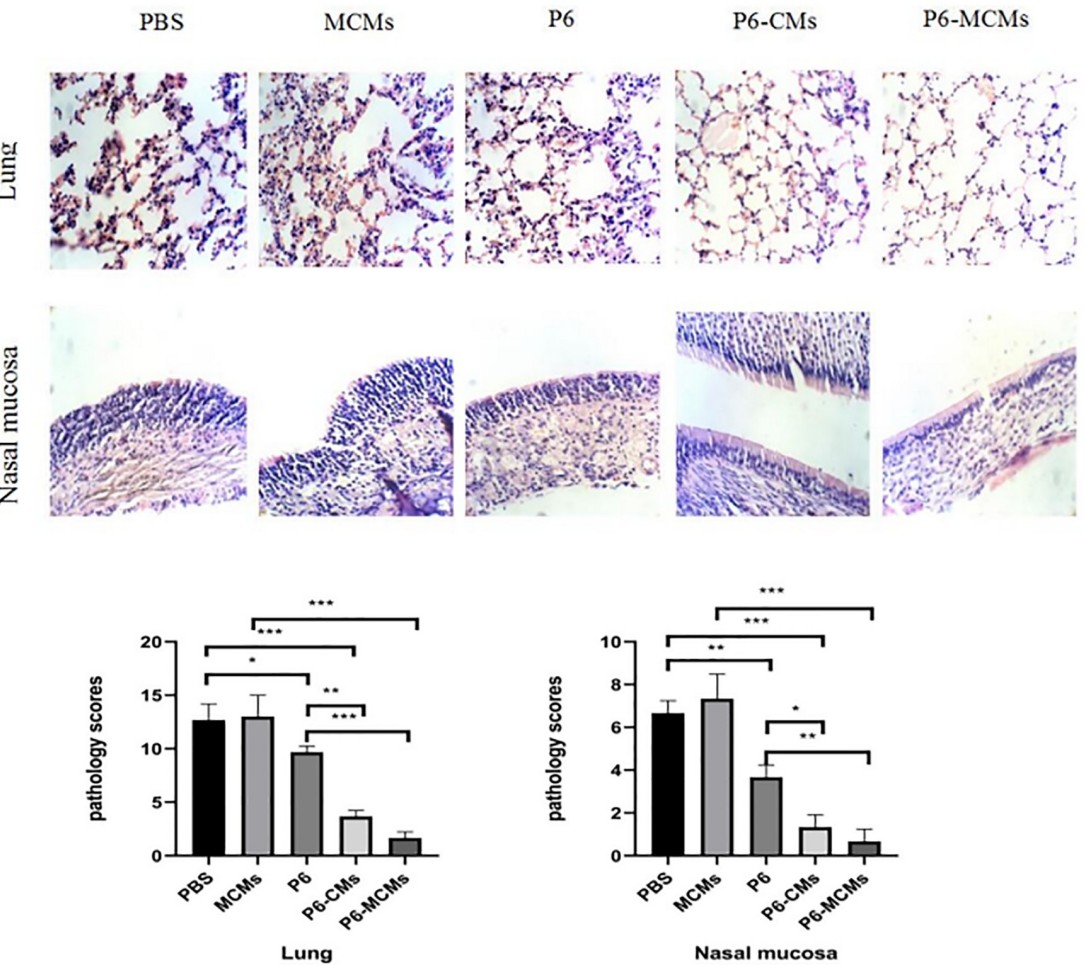

**Fig 7. Hematoxylin-Eosin staining of the nasal mucosa and lung tissues to evaluate histopathologic alterations in mice.** Histological scores for lung, nasal mucosa tissues from mice (n = 3 per group). $^*P<0.05$, $^{**}P<0.01$ and $^{***}P<0.001$.

### Protection against nontypeable *Haemophilus influenzae* infection

Nasal mucosa and lung tissues were collected, and histopathologic examination was performed by hematoxylin-eosin staining one week after intranasal inoculation with nontypeable *Haemophilus influenzae*. As follow the Fig 7, in the PBS and MCM groups, the reticular structure of the lung tissue was damaged, inflammatory cells were increased, the cilia of the nasal mucosa were disordered, with lodging, and some even disappeared, the columnar epithelial cells were loosely arranged, and lymphocyte and inflammatory cells were increased in the lamina propria; In the P6-MCM group, the nasal mucosa and lung tissues had a mild influx of inflammatory cells, in contrast to the PBS and MCM groups.

### Discussion

NTHi is a conditional pathogen that colonizes the human nasopharynx. It can cause secondary infections when the body's resistance is low or the local microecological environment is unbalanced, such as in childhood OM, cystic fibrosis, community-acquired pneumonia, and chronic common infections such as bronchitis, conjunctivitis, and acute exacerbation of COPD in

adults [7, 8]. Among them, the acute exacerbation of COPD in children and adults worldwide has the highest incidence [2]. With the development of vaccine adjuvants and carriers, we used mannose-modified chitosan microspheres to load the NTHi recombinant outer membrane protein P6 for the first time, and it showed that the microsphere vaccine can mightily weaken the invasion of NTHi to lung tissue and nasal mucosa tissue.

Although NTHi has been studied for many years worldwide, we are still unable to effectively control and prevent its infection. Studies have found that the NTHi outer membrane protein plays an important role in its infection, pathogenicity and interaction with host cells, which lead to host disease. Protective immunity and several potentially advantageous outer membrane proteins, P2, P5, P6, protein D, protein E, and Haps protein, have been listed as NTHi vaccine candidates. Among them, the P6 protein is highly conserved and passes through the mouse nasal mucosa. Immunization can induce the production of high titers of specific mIgA and IgG antibodies, induce spleen CD4 T cells to express P6-specific Th2 cytokines [33], and stimulate the body's immune protection and the clearance of NTHi [33, 34].

Chitosan (CS) is a high-molecular-weight polysaccharide that can be slowly degraded to nontoxic glucosamine by lysosomal enzymes and then completely absorbed by the body. It has been used as a mucosal particle carrier for a variety of vaccines and drugs. In this study, the ion cross-linking method was used to cross-link the negative charge of the anionic coagulant TPP and the positive charge of the primary amino group in chitosan to form spherical particles at a suitable pH and the mass ratio of chitosan:TPP. Studies have found that a suitable TPP concentration has a strong effect on the formation of microspheres; a smaller TPP concentration is not conducive to the formation of microspheres, and a larger TPP concentration will cause chitosan to form flocculent precipitate. Bodmeier et al. [35] and Kubiak [36] have Reported that when the pH of the solution is 4~6 and the mass ratio of chitosan and TPP is between 3:1 and 6:1, it is possible to form more stable spherical particles. For the formation of uniform spheres, adding the appropriate dispersant Tween-80 can reduce adhesion. The successful loading of P6 protein on CS microspheres provides protection and reduces the degradation of P6 before it reaches the target site. Studies such as that performed by Wu et al. [37] showed that CS nanoparticles loaded with the natural anticancer drug Res can effectively retain high antioxidant and anticancer activity and improve stability, solubility and tumor targeting. The mucosal adhesion and high permeability of the CS microsphere carrier facilitate the absorption of P6 protein by the nasal mucosa and minimize the loss of protein. Studies have shown that oral administration of an insulin-loaded CS microsphere vaccine in diabetic rats can effectively enhance the absorption of insulin by the intestinal mucosa and improve bioavailability [38]. CS microspheres can also effectively target nasal mucosa-related lymphoid tissues, enhance specific immune responses, and increase the levels of IgG and IgA antibodies [39–41].

Chitosan is obtained by the deacetylation reaction of chitin. It is only soluble in dilute acid but insoluble in water and organic solvents, which increases the difficulty of chitosan research to a certain extent. In recent years, the development of chitosan derivatives such as carboxymethyl chitosan, quaternary ammonium chitosan and N-succinyl chitosan has improved the water solubility of chitosan. Mannose is currently the only glyconutrient used in the clinic. It can be used to directly synthesize glycoproteins. The hydrophilic mannose-modified mannose derivatives have high biocompatibility. The microsphere carrier can target antigen extraction. Receptor endocytosis mediated by the mannose receptor on the presenting cell improves antigen presentation and enhances the immune response. In this study, mannose was covalently combined with chitosan under the action of sodium cyanoborohydride by reductive amination. The results showed that the ratio of glacial acetic acid to methanol and the amount of catalyst used will affect the free amino groups of chitosan. If the degree of substitution is too

large, it is not conducive to the formation of spheres. Studies have found that when the degree of substitution of free amino groups in chitosan is 5% to 30%, the formation of microspheres will not be affected [18, 19]. It can be seen from Table 2 that the zeta potential of the modified chitosan microspheres is reduced, and it is easier to couple targeting molecules on the surface of the modified chitosan microspheres, which greatly increases the protein loading on the surface so that a small amount of microsphere carriers can achieve the same immune effect, thereby saving money and reducing the immune dosage. Moreover, the structure of the modified chitosan microspheres changes, which speeds up the release of proteins.

Regulating immunity, especially facilitating more effective antigen presentation by antigen-presenting cells (APCs) and activating immune effector T cells and B cells, is the main goal for the treatment and prevention of bacterial or viral infections, as well as the development of efficient vaccines. The mannose receptor is widely expressed on the surface of APC cells. The mannose-modified chitosan microsphere carrier can target the mannose receptor and can be more effectively presented. Zhu et al. [42], Jiang HL. [19] and Cui Z. [20] proved that chitosan microspheres modified with mannose can target the mannose receptor on the surface of mouse RAW264.7 macrophage-like cells in vitro. The mucosa of the upper respiratory tract is the invasion pathway of many bacterial viruses, such as NTHi. It is very important to set up biological barriers on the surface of mucosa. The mucosal antibody IgA plays a dominant role in the mucosal barrier. This study shows that after immunization via the nasal mucosa, the levels of serum IgG and mucosal IgA in the P6-MCM group were significantly increased compared to those in the P6-CM group, indicating that the microsphere carrier not only enhanced mucosal immunity but also formed a defensive wall on the mucosal surface. This system can enhance humoral immunity and weaken mightily the invasion of NTHi.

Th cells play a central role in the cellular immune response. It can assist B cells in producing antibodies, activate macrophages to kill intracellular antigens, and promote the formation of CTLs. Th1 cells can assist cellular immunity; Th2 cells can assist humoral immunity; and Th17 cells can induce autoimmunity, activate neutrophils, guide Th1 cells to the bacterial replication site and participate in protective immunity against intracellular infection. In this experimental study, Th1 and Th2 cytokines were detected, and it was found that the levels of IL-2, IFN-γ, IL-4 and IL-5 cytokines in the P6-MCM group were significantly increased, while in the P6-CM group, only IL- 4 levels were significantly increased, indicating that the chitosan microsphere vaccine maybe mainly stimulate a Th2-type cellular immune response, while the mannose-modified microsphere vaccine can stimulate both Th2-type and Th1-type cellular immune responses, that is, a mixed Th1/Th2 cellular immune response. It was found that Th17 cytokine levels in the P6-CM and P6-MCM groups were significantly increased, indicating that both agents can promote the differentiation of Th cells into Th17 cells. Through experimental research on the proliferation of spleen lymphocytes, it was shown that the mannose-modified microsphere vaccine is not only effective in stimulating an immune response but also involved in the T cell proliferation stage. It is worth noting that the adjuvants and antigen delivery systems currently studied worldwide are mainly exogenous antigens that enter the MHC class II-restricted presentation pathway and induce antibody-mediated immune responses. For therapeutic vaccines, it is mostly necessary to initiate cellular immune responses. There is a need for an endogenous presentation pathway restricted by MHC class I molecules that deliver the antigen to the cell. In this study, flow cytometry was used to detect the subpopulation ratio of spleen lymphocytes in immunized mice. The ratio of CD3+CD4 + and CD3+CD8+ T cells was significantly higher in the P6-MCM group than in the other groups. The endogenous presentation pathway is restricted by MHC class I molecules; that is, the mannose-modified microsphere vaccine maybe also undergo endocytosis mediated by the mannose receptor and induce MHC class I-restricted immune activation. It presents a way to

stimulate both the Th cell immune response mediated by CD4+ T cells and CTL killing mediated by CD8+ T cells. Wu et al. [43] used MCMs loaded with Mycobacterium pulmonary nucleic acid DNA to prepare a tuberculosis vaccine. After immunization, this vaccine also induced a Th1 cellular immune response in mice and activated lung tissue CD4+ and CD8+ T cell immune responses. Chieppa et al. [23] and Cui et al. [20] used MCMs to load *Pseudomonas aeruginosa* outer membrane protein OprF190–342-OprI21–83. After immunization of mice, MCMs caused a mixed Th1/Th2 cellular immune response and CD8+ T cell-mediated CTL-based immunity. In addition, Wilk and Mills [44] have found that vaccination can also produce tissue-resident memory T (TRM) cells, which play a vital role in maintaining long-term protective immunity against mucosal pathogens, especially a vaccine that produces Th1 and Th17 reactions. Therefore, the P6 protein microsphere vaccine modified by mannose maybe also promote the formation of TRM cells and cause strong mucosal immunity, while it need further studies and more evidence to demonstrate this hypothesis.

In the protective immunity experiment, the nasal mucosa and lung tissue of the control group and the P6 group showed pathological changes, while the tissues of the microsphere vaccine group had a mild influx of inflammatory cells, in contrast to the PBS and MCM groups, especially in the mannose-modified group, which exhibited stronger immune protection and weaken mightily the invasion of NTHi. In this experiment, a nontypeable *Haemophilus influenzae* microsphere vaccine was successfully prepared, and animal experiments showed that the vaccine can provide strong protection against NTHi infection. However, it remains unclear how the mannose-modified microsphere P6 protein vaccine carries out MHC class I endogenous presentation through targeted receptors. The mechanism and whether this vaccine promotes the production of TRM cells still needs further study based on experimental evidence.

## Supporting information

**S1 Raw images.**
(DOCX)

## Author Contributions

**Conceptualization:** Yushuai Ma, Yutuo Zhang.

**Data curation:** Ying Zhao, Yutuo Zhang.

**Funding acquisition:** Yutuo Zhang.

**Investigation:** Yushuai Ma, Yutuo Zhang.

**Methodology:** Yushuai Ma, Ying Zhao, Rui Chen.

**Project administration:** Yushuai Ma, Yutuo Zhang.

**Resources:** Yushuai Ma, Ying Zhao, Yanxia Zhang, Haixia Qiao, Yutuo Zhang.

**Software:** Yushuai Ma, Ying Zhao, Wanru Sun, Yueli Chang, Shaoping Kang.

**Supervision:** Yutuo Zhang.

**Validation:** Yutuo Zhang.

**Visualization:** Yushuai Ma, Ying Zhao.

**Writing – original draft:** Yushuai Ma.

**Writing – review & editing:** Yutuo Zhang.

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
