## [Decision Letter · Decision Letter 0]

30 Mar 2021

PONE-D-21-06673

Mucosal immunity of mannose-modified chitosan microspheres loaded with the nontypable Haemophilus influenzae outer membrane protein P6 in BALB/c mice

PLOS ONE

Dear Dr. yutuo,

Thank you for submitting your manuscript to PLOS ONE. After careful consideration, we feel that it has merit but does not fully meet PLOS ONE’s publication criteria as it currently stands. Therefore, we invite you to submit a revised version of the manuscript that addresses the points raised during the review process.

Two expert reviewers appreciated the topic of the study.  However, they raised a number of concerns regarding the methodologies used, the sub-optimal description of methodologies, group sizes etc., which make it difficult for a reviewer to gauge the validity of interpretations described in the manuscript.  Additionally, the reviewers raise questions regarding the short time interval between immunization and challenge, and whether any immune response analyses were conducted after challenge.  I believe addressing of all these points will be key and I urge you to do so, should you decide to resubmit a revised version of the manuscript.

We look forward to receiving your revised manuscript.

Kind regards,

Ashlesh K Murthy, M.D., Ph.D.

Academic Editor

PLOS ONE

Journal Requirements:

2. To comply with PLOS ONE submissions requirements, in your Methods section, please provide additional information on the animal research and ensure you have included details on (1) methods of sacrifice, (2) methods of anesthesia and/or analgesia, and (3) efforts to alleviate suffering.

3. Thank you for including your ethics statement:  "I indicate that the animal reseaarch ethics committe prospectively approved this research, and the BALB/c mice were cared for in the Laboratory Animal Center of Life Science Research Center (Hebei North University, Zhangjiakou, China),".   

Please amend your current ethics statement to include the full name of the ethics committee that approved your specific study.

For additional information about PLOS ONE submissions requirements for ethics oversight of animal work, please refer to http://journals.plos.org/plosone/s/submission-guidelines#loc-animal-research  

Reviewers' comments:

Reviewer's Responses to Questions

**Comments to the Author**

1. Is the manuscript technically sound, and do the data support the conclusions?

Reviewer #1: Partly

Reviewer #2: Partly

2. Has the statistical analysis been performed appropriately and rigorously? 

Reviewer #1: Yes

Reviewer #2: I Don't Know

3. Have the authors made all data underlying the findings in their manuscript fully available?

Reviewer #1: Yes

Reviewer #2: Yes

4. Is the manuscript presented in an intelligible fashion and written in standard English?

Reviewer #1: Yes

Reviewer #2: Yes

5. Review Comments to the Author

Reviewer #1: The manuscript entiteled “Mucosal immunity of mannose-modified chitosan microspheres 1 loaded with the 2 nontypable Haemophilus influenzae outer membrane protein P6 in BALB/c mice” aims to describe the use of MCM as a delivery system for P6 as a potential vaccine construct against NTHi. While the data presented in this manuscript in relevant to the field, the methods lack detail. Further, some findings are overstated and are purely speculative based on results from other studies. My comments are as follows:

1. Line 17, 48, 330 – Nontypeable

2. Statement starting on line 53 – I feel this is an over statement of the effectiveness of the vaccine. I would consider rewording as the vaccine effectiveness reportedly is variable in some infants, some age groups, or some forms (infection vs. invasive disease). While the initial impact of the vaccine was impressive when it was released, there has been an increase in Hib infections over the years despite vaccination control programs.

3. Only female mice were utilized, was the impact of gender on immune responsiveness to the vaccine considered. This should be cited as a limitation of the study.

4. Line 176 – how were T cells isolated beyond the use of LSM? If only LSM was used how can you be sure that the response that was evaluated was a T cell response and not a result of other contaminant cells (i.e. B cells, macrophages, DC)? What was the viability of these cells?

5. Line 184 – same comment as above. How can you be confident that division was T cells only when only LSM was utilized? What was the viability of these cells?

6. A complete description briefly describing the separation protocol and how lavage and blood samples were obtained should be included or at a minimum referenced. How were the mice sacrificed for histopathology?

7. Line 192 – spleen

8. State in the methods when following vaccination, the mice were inoculated with NTHi.

9. Which data were analyzed utilizing an unpaired T test?

10. Line 282 – confusing as written. Consider “at the P6 or MCM group…”

11. Fig 5B include what IFN in the title of the graph.

12. Paragraph starting on line 300 should be reworded based on response to the above described concerns with the methods. Further the use of the word “efficient” seems like an overstatement Based on the appearance of the graph the biological significance of this result would be questionable.

13. Did a pathologist review the tissue slides or who provided the changes described in the manuscript? What is meant by the worded “tended” to be normal? How many mice were utilized for this experiment? At what magnification were these images from?

14. Sentence starting on line 346 is overstated as antigen uptake and presentation were not directly assessed in these experiments. Further these results did not t show that P6-CM or P6-MCM “prevented” infection as infection was not assessed. Perhaps you are referring to evidence of disease based on histopathologic evaluation?

15. Reference for line 360?

16. What is meant by “deploy” the mucosa (line 411)

17. Sentence starting on line 416 is overstated as colonization or possible invasion were not evaluated herein.

18. Sentence staring on line 423 based on figure 5 I disagree that P6-CM “only” induces a Th2 response. Maybe the magnitude of production was greatest for IL-4, P6-CM did in fact induce production of the other cytokines to varying degrees. What were the p-values for the other cytokines for P6-CM vs. controls? The immune response to most antigens is a mixed response but in some cases one type of response predominates which I agree may be the case for P6-CM. Consider re-wording.

19. Line 434 – what is meant by foreign? Do you mean exogenous? What about cross-presentation?

20. Sentence staring on line 438 – this is speculative, this may have happened I agree but was not assessed in these experiments.

21. Line 454 and Line 457 overstated please reword.

22. Line 460 prevention of infection was not assessed. Please remove.

Reviewer #2: This study by Zhang et al titled “ Mucosal immunity of mannose-modified chitosan microspheres loaded with the nontypable Haemophilus influenzae outer membrane protein P6 in BALB/c mice” studied the mannose-modified chitosan (MC) microspheres as a adjuvant-delivery system for vaccines against NTHi. Zhang et al used NTHi outer membrane protein P6, as a potential vaccine candidate and observed after intranasal immunization P6-MCMs enhance both systemic and mucosal immune responses. They also observed that P6-MCMs enhance cellular immunity and trigger a mixed Th1/Th2-type immune response and also Th17-type immune response. There are some questions that needs to be answered

1. Authors didn’t mentioned how many mice were used in each group?

2. All the studies were conducted two weeks after last vaccination. Were these studies done at later time points?

3. In the Figure 4 where IgA levels were measured in the Nasal and lung lavage fluids, were the samples pooled or measured individually?

4. From the graphs the difference between the groups looks minimal. Is it possible to re run the statistics please?

5. After final vaccination mice were challenged with nontypable Haemophilus influenzae tissues were collected and histopathologic examination was done. Were any other studies were done besides histopathologic examination?

6. PLOS authors have the option to publish the peer review history of their article (what does this mean?). If published, this will include your full peer review and any attached files.

Reviewer #1: No

Reviewer #2: No

---

## [Author Response · Author response to Decision Letter 0]

13 May 2021

Reviewer #1:

1 The concerns regarding the methodologies used, the sub-optimal description of methodologies, group sizes.?

Answers: The date of study groups with control group were analyzed utilizing an unpaired T test, such as P6 with PBS, P6-CMs with PBS, P6-MCMs with PBS, P6-MCMs with MCMs. While the compare among study groups P6, P6-CMs, P6-MCMs were analyzed by ANOVA test. 95 mice total were contributed for this study, 15 mice every group, 15 mice candidate. The available date of 3 mice every group were analysed statistically for antibody, cytokines, lymphocyte proliferation and T lymphocyte subpopulation assay. 4 mice every group for histopathologic examination.

2. The questions regarding the short time interval between immunization and challenge, and whether any immune response analyses were conducted after challenge.?

Answers: Immunization and challenge time were based on our Laboratory animal protocol, but we will try to extend immunization and challenge time according to your kindly suggestion. Furthermore , I’m so sorry that I have no time to conduct any immune response analyses after challenge to prove the effectiveness of the vaccine from multiple angles because of the graduation date, so that this should be a limitation of the study.

3. Additional information on the animal research and ensure you have included details on (1) methods of sacrifice, (2) methods of anesthesia and/or analgesia, and (3) efforts to alleviate suffering.?

Answers: Mice were anesthetized via intraperitoneal injection of sodium pentobarbital (1%, 50 mg/kg) and sacrificed via rapid dislocation of the necks. All efforts were made to minimize animal suffering and to reduce the number of animals used.

4. Only female mice were utilized, was the impact of gender on immune responsiveness to the vaccine considered. This should be cited as a limitation of the study.?

Answers: In this study, only female mice were used, merely considering that female mice usually live in peace, they are easier to raise in groups than male mice.

5. How were T cells isolated beyond the use of LSM? If only LSM was used how can you be sure that the response that was evaluated was a T cell response and not a result of other contaminant cells.?

Answers: The cytokines in splenic lymphocytes were measured, lymphocytes were isolated from spleen tissue with lymphocyte separation medium. The same as the splenic lymphocyte proliferation assay.

6. A complete description briefly describing the separation protocol and how lavage and blood samples were obtained should be included or at a minimum referenced. How were the mice sacrificed for histopathology.?

Answers: Blood samples were collected via the retro-orbital sinuses into drop tubes. The collection methods of lavage are as follows: The mice were sacrificed via rapid dislocation of the necks and dissected subsequently, the trachea was exposed and ligated from the middle section. After then, 500 �l PBS was injected in the trachea toward the lungs, thus the lungs lavage were obtained from the trachea after 5 minutes with gently kneading lung. Injecting 500 �l PBS toward the nasopharynx the same way, collecting the fluid flowing out of the nasal cavity, thus the nasal cavity lavage were obtained after repeating three times.

7. State in the methods when following vaccination, the mice were inoculated with NTHi.?

Answers: Mice vaccination and inoculation with NTHi were all performed via intranasal drip. the mice were anesthetized and challenged with NTHi (ATCC 49247) in a bacterial suspension containing an dose (LD100) (1×108 CFU/ml) via intranasal drip via intranasal drip after the last immunization

8. Which data were analyzed utilizing an unpaired T test.?

Answers: The date of study groups and control group were analyzed utilizing an unpaired T test, such as P6 and PBS, P6-CMs and PBS, P6-MCMs and PBS, P6-MCMs andMCMs

9. Paragraph starting on line 300 should be reworded based on response to the above described concerns with the methods. Further the use of the word “efficient” seems like an overstatement Based on the appearance of the graph the biological significance of this result would be questionable.

 Answers: From the experimental data, the use of the word “efficient” seems like an overstatement, in addition, anyother immune response analyses were not conducted after challenge with bacteria, so that there lack of sufficient evidence to confirm the “efficient” of vaccine. We reword the word “efficient” as “effective”.

9. Did a pathologist review the tissue slides or who provided the changes described in the manuscript? What is meant by the worded “tended” to be normal? How many mice were utilized for this experiment? At what magnification were these images from?

Answers: We have consulted the pathologist in North University of Hebei for the histopathological changes. “tended to be normal” reworded “the nasal mucosa and lung tissues had a mild influx of inflammatory cells”. 4 mice every group for histopathologic examination, and 40 times magnification for these images.

10. Sentence starting on line 346 is overstated as antigen uptake and presentation were not directly assessed in these experiments. Further these results did not t show that P6-CM or P6-MCM “prevented” infection as infection was not assessed. Perhaps you are referring to evidence of disease based on histopathologic evaluation?

Answers: I’m so sorry the I have no time to conduct any immune response analyses after challenge to prove the effectiveness of the vaccine against NTHi from multiple angles. The sentences of antigen uptake and presentation were removed and reworded accordingly. But it showed that the microsphere vaccine can really reduce the invasion of NTHi to lung tissue and nasal mucosa tissue.

11. What is meant by “deploy” the mucosa (line 411)?

Answers: It reworded as “set up biological barriers on the surface of mucosa.”

12. Sentence starting on line 416 is overstated as colonization or possible invasion were not evaluated herein.

 Answers: It removed the sentence “to prevent NTHi from invading”. And reworded “eliminate the invaded NTHi” as “weaken the invasion of NTHi”

13. Sentence staring on line 423 based on figure 5 I disagree that P6-CM “only” induces a Th2 response. Maybe the magnitude of production was greatest for IL-4, P6-CM did in fact induce production of the other cytokines to varying degrees. What were the p-values for the other cytokines for P6-CM vs. controls? The immune response to most antigens is a mixed response but in some cases one type of response predominates which I agree may be the case for P6-CM. Consider re-wording.?

Answers: It reworded as “the chitosan microsphere vaccine maybe mainly stimulate a Th2-type cellular immune response”. We also add the p-values for the other cytokines for P6-CM vs. controls on figure 5. 

Reviewer #2:

1. Authors didn’t mentioned how many mice were used in each group?

Answers: 95 mice total were contributed for this study, 15 mice every group, 15 mice candidate. The available date of 3 mice every group were analysed statistically for antibody, cytokines, lymphocyte proliferation and T lymphocyte subpopulation assay. 4 mice every group for histopathologic examination.

2. All the studies were conducted two weeks after last vaccination. Were these studies done at later time points?

Answers: All the studies were conducted two weeks after last vaccination and done at later time points.

3. In the Figure 4 where IgA levels were measured in the Nasal and lung lavage fluids, were the samples pooled or measured individually?

Answers: IgA levels were measured in the Nasal and lung lavage fluids individually.

5. After final vaccination mice were challenged with nontypable Haemophilus influenzae tissues were collected and histopathologic examination was done. Were any other studies were done besides histopathologic examination?

Answers: I’m so sorry that I have no time to conduct any immune response analyses after challenge to prove the effectiveness of the vaccine from multiple angles because of the graduation date, so that this should be a limitation of this study.

---

## [Decision Letter · Decision Letter 1]

28 Jul 2021

PONE-D-21-06673R1

Mucosal immunity of mannose-modified chitosan microspheres loaded with the nontypable Haemophilus influenzae outer membrane protein P6 in BALB/c mice

PLOS ONE

Dear Dr. yutuo,

Thank you for submitting your manuscript to PLOS ONE. After careful consideration, we feel that it has merit but does not fully meet PLOS ONE’s publication criteria as it currently stands. Therefore, we invite you to submit a revised version of the manuscript that addresses the points raised during the review process.

Both reviewers raise a number of concerns regarding methodologies and controls being insufficient to draw reasonable conclusions.  I urge you to provide additional evidence to support your interpretations, as I cannot accept manuscript for publications based on the fact that there was insufficient time before graduation.

We look forward to receiving your revised manuscript.

Kind regards,

Ashlesh K Murthy, M.D., Ph.D.

Academic Editor

PLOS ONE

Journal Requirements:

Additional Editor Comments (if provided):

Reviewers' comments:

Reviewer's Responses to Questions

**Comments to the Author**

1. If the authors have adequately addressed your comments raised in a previous round of review and you feel that this manuscript is now acceptable for publication, you may indicate that here to bypass the “Comments to the Author” section, enter your conflict of interest statement in the “Confidential to Editor” section, and submit your "Accept" recommendation.

Reviewer #2: (No Response)

Reviewer #3: (No Response)

2. Is the manuscript technically sound, and do the data support the conclusions?

Reviewer #2: Partly

Reviewer #3: Partly

3. Has the statistical analysis been performed appropriately and rigorously? 

Reviewer #2: Yes

Reviewer #3: Yes

4. Have the authors made all data underlying the findings in their manuscript fully available?

Reviewer #2: Yes

Reviewer #3: Yes

5. Is the manuscript presented in an intelligible fashion and written in standard English?

Reviewer #2: Yes

Reviewer #3: No

6. Review Comments to the Author

Reviewer #2: This study by Zhang et al titled “ Mucosal immunity of mannose-modified chitosan microspheres loaded with the nontypable Haemophilus influenzae outer membrane protein P6 in BALB/c mice” studied the mannose-modified chitosan (MC) microspheres as a adjuvant-delivery system for vaccines against NTHi. Zhang et al used NTHi outer membrane protein P6, as a potential vaccine candidate and observed after intranasal immunization P6-MCMs enhance both systemic and mucosal immune responses. They also observed that P6-MCMs enhance cellular immunity and trigger a mixed Th1/Th2-type immune response and also Th17-type immune response.

The effectiveness of mannose-modified chitosan microspheres loaded with outer membrane protein P6 in BALB/c mice is not conclusive. Besides, from histopathologic examination of slides there isn’t any conclusive evidence.

1. Did the authors compared bacterial burden between the groups after infection?

2. Were Cytokine Levels in Spleen Lymphocyte Culture Supernatants and T Lymphocyte Proliferation Assays compared between groups after infection?

3. Is there any pathological scoring of tissues performed?

4. Please correct sleep in line 207 to spleen

5. In line 217 via intranasal drip is repeated twice

6. In line 433 central is misspelled

Reviewer #3: The current paper by Ma Y and group, proposes mannose-modified chitosan microsphere-based delivery system to increase immunization efficiency against Haemophilus influenzae using outer membrane protein P6 as antigen. The authors have cloned, expressed and purified P6 antigen, which is then loaded into either mannose-modified chitosan or unmodified chitosan microspheres. The authors report enhanced immune response elicited by the modified matrix delivery system (both systematic and mucosal response) in comparison with the unmodified counterpart. The modified matrix assisted immunized mice show increased levels of IgG and IgA in serum and lavage fluid respectively. In addition, the modified matrix also enhanced the response of both humoral and cellular immunity response. Further, authors challenged the immunized mice with the bacterial suspension and observed the integrity of the lungs was maintained in the mice that received modified matrix P6 immunization, with lower infiltration of immune cells.

Some of the major concerns about the paper are:

1. As a proof principle, the study lacks details on generation of microspheres. The authors argue that mannose modification of chitosan helps in better uptake of the microsphere by the immune cells. Since, the modification is “homemade” at about 19% efficiency, it is not clear if the desired uptake is achieved or not. Do the authors enrich modified chitosan before microspheres generation? Did the authors check if the cellular uptake of modified chitosan was increased in vitro? Do they achieve similar degree of immune response if a commercial (modified) version is used (if available)?

2. How consistent is the dimension of the microspheres formed? It is not clear how many times the microspheres were consistently generated and if each time it elicited similar degree of immune response.

3. Are there any positive controls that the authors have used to see the efficiency of the immune response using the modified matrix?

4. In Figure 7 experiment, how do the authors differentiate loss of integrity because of active infection and primed defense response by immune cells? If the authors use heat killed or dead bacteria, do they see similar levels of immune cell infiltration in the mice? How does the normal infected lung look?

7. PLOS authors have the option to publish the peer review history of their article (what does this mean?). If published, this will include your full peer review and any attached files.

Reviewer #2: No

Reviewer #3: No

---

## [Author Response · Author response to Decision Letter 1]

11 Sep 2021

Reviewer #2:

1. Did the authors compared bacterial burden between the groups after infection?

Answers: When the experiment was originally designed, it did include the bacterial burden in the mouse nasal cavity, but later due to the difficultly available NTHI in the nasal cavity and various operational reasons, the bacterial load experiment failed to proceed as scheduled.

2. Were Cytokine Levels in Spleen Lymphocyte Culture Supernatants and T Lymphocyte Proliferation Assays compared between groups after infection?

Answers: We only measure the cytokines in the spleen lymphocyte culture supernatants, while not compare the cytokine levels in the splenic lymphocyte culture supernatants and the T lymphocyte proliferation assays between the groups. We will take your suggestions seriously and design experiments in the future. It can further confirm the changes in cytokine secretion after cell proliferation.

3. Is there any pathological scoring of tissues performed??

Answers: When the results of the experiment came out, I consulted a professional pathologist to observe the obvious changes in inflammatory cell infiltration between the groups, especially in the degree of congestion and destruction of the lung tissue network structure, and cilia lodging and defect in the nasal mucosa group. These obvious changes can be observed with the naked eye, so we don’t have performed pathological scoring of tissues

Reviewer #3:

1. As a proof principle, the study lacks details on generation of microspheres. The authors argue that mannose modification of chitosan helps in better uptake of the microsphere by the immune cells. Since, the modification is “homemade” at about 19% efficiency, it is not clear if the desired uptake is achieved or not. Do the authors enrich modified chitosan before microspheres generation? Did the authors check if the cellular uptake of modified chitosan was increased in vitro? Do they achieve similar degree of immune response if a commercial (modified) version is used (if available)?

Answers: The detailed methods have been presented in lines 140-150, including the ratio of chitosan and TPP, PH of solutions, and the reaction time. Microspheres were prepared by the ionotropic gelation process following the report of Jiang et al and optimized. If the degree of substitution is too large, it is not conducive to the formation of spheres. Studies have found that when the degree of substitution of free amino groups in chitosan is 5% to 30%, the formation of microspheres will not be affected. It is stated that the method of loading P6 antigen with mannose-modified chitosan microspheres can enhance the P6 immune response. As for the degree of substitution of mannose modification that can maximize the efficiency of the mannose receptor method, we have not studied too much. We did not carry out the process of mannose targeting the mannose receptor in vitro, which is also a defect of this article, but it can infer that MCMs was captured through receptor-mediated endocytosis by targeting MR, which improves the antigen uptake efficiency of APC according the experimental data, so that enhanceing the efficiency of antigen presentation.

2. How consistent is the dimension of the microspheres formed? It is not clear how many times the microspheres were consistently generated and if each time it elicited similar degree of immune response.?

Answers: The uniformity of the formed microspheres was good, and the distribution of the microspheres was measured. It shows the distribution of the diameter of the microspheres by intensity in Fig 3. The formation method of the microspheres is the reference Microspheres were prepared by the ionotropic gelation process following the report of Jiang et al [31] and slightly optimized, it can ensure the reproducibility of formation in vitro.

3. Are there any positive controls that the authors have used to see the efficiency of the immune response using the modified matrix?

Answers: The experimental components of this experiment are 5 groups, PBS, MCMs, P6, P6-CMs, P6-MCMs. Among them, the MCMs group can serve as positive controls.

4. In Figure 7 experiment, how do the authors differentiate loss of integrity because of active infection and primed defense response by immune cells? If the authors use heat killed or dead bacteria, do they see similar levels of immune cell infiltration in the mice? How does the normal infected lung look??

Answers: We really did not compare the levels of immune cell infiltration in the mice between the aggressive bacteria and dead bacteria. Heat killed or dead bacteria does not have the ability to actively attack, and it is probably different from the extent of damage to tissues caused by aggressive bacteria. It can be clearly seen that the histopathological improvement of mice immunized with microspheres, the lung tissues have obvious bleeding, inflammatory cell infiltration, and tissue texture destruction in normal group.

---

## [Decision Letter · Decision Letter 2]

5 Oct 2021

PONE-D-21-06673R2Mucosal immunity of mannose-modified chitosan microspheres loaded with the nontypable Haemophilus influenzae outer membrane protein P6 in BALB/c micePLOS ONE

Dear Dr. yutuo,

Thank you for submitting your manuscript to PLOS ONE. After careful consideration, we feel that it has merit but does not fully meet PLOS ONE’s publication criteria as it currently stands. Therefore, we invite you to submit a revised version of the manuscript that addresses the points raised during the review process.

As you will see in the reviewer comments, the revised version of the manuscript fails to address the prior comments raised by the reviewers.  The data set and methodologies as written do not allow for a critical review of the findings.  As such, I urge you to consider submitting a revised version of the manuscript only if significant new changes, including methodological changes and new data acquired using such methodologies can be presented.

We look forward to receiving your revised manuscript.

Kind regards,

Ashlesh K Murthy, M.D., Ph.D.

Academic Editor

PLOS ONE

Reviewers' comments:

Reviewer's Responses to Questions

**Comments to the Author**

1. If the authors have adequately addressed your comments raised in a previous round of review and you feel that this manuscript is now acceptable for publication, you may indicate that here to bypass the “Comments to the Author” section, enter your conflict of interest statement in the “Confidential to Editor” section, and submit your "Accept" recommendation.

Reviewer #2: (No Response)

Reviewer #3: (No Response)

2. Is the manuscript technically sound, and do the data support the conclusions?

Reviewer #2: Yes

Reviewer #3: No

3. Has the statistical analysis been performed appropriately and rigorously? 

Reviewer #2: Yes

Reviewer #3: I Don't Know

4. Have the authors made all data underlying the findings in their manuscript fully available?

Reviewer #2: Yes

Reviewer #3: Yes

5. Is the manuscript presented in an intelligible fashion and written in standard English?

Reviewer #2: Yes

Reviewer #3: No

6. Review Comments to the Author

Reviewer #2: This study by Zhang et al titled “ Mucosal immunity of mannose-modified chitosan microspheres loaded with the nontypable Haemophilus influenzae outer membrane protein P6 in BALB/c mice” studied the mannose-modified chitosan (MC) microspheres as a adjuvant-delivery system for vaccines against NTHi. Zhang et al used NTHi outer membrane protein P6, as a potential vaccine candidate and observed after intranasal immunization P6-MCMs enhance both systemic and mucosal immune responses. They also observed that P6-MCMs enhance cellular immunity and trigger a mixed Th1/Th2-type immune response and also Th17-type immune response.

In response to the comments from the previous revision regarding pathological scoring, authors mentioned that there is a clear visual difference between the groups and hence scoring is not performed. I would like the authors to do scoring done by an independent pathologist to avoid Experiment bias and make a graph to show the difference.

Reviewer #3: (No Response)

7. PLOS authors have the option to publish the peer review history of their article (what does this mean?). If published, this will include your full peer review and any attached files.

Reviewer #2: No

Reviewer #3: No

---

## [Author Response · Author response to Decision Letter 2]

18 Nov 2021

Differences between groups were analyzed by pathology scores[32]. To score lung inflammation and damage, the following parameters: edema, interstitial inflamamation, intra-alveolar inflammation, endothelialitis, hyperemia, degree of inflammatory cell infiltration. Each parameter was graded from 0 (absent) to 4 (severe). Nasal mucosa were scored according to the following parameters: presenc and degree of inflammation cell infiltration, presence and degree of the cilia of the nasal mucosa disappearing, degree of looseness of the columnar epithelial cells arranging. Each parameter was graded from 0 (absent) to 3 (severe). The total pathology scores were expressed as the sum of the score for all parametrs.

---

## [Decision Letter · Decision Letter 3]

9 Feb 2022

PONE-D-21-06673R3Mucosal immunity of mannose-modified chitosan microspheres loaded with the nontypable Haemophilus influenzae outer membrane protein P6 in BALB/c micePLOS ONE

Dear Dr. yutuo,

Thank you for submitting your manuscript to PLOS ONE. After careful consideration, we feel that it has merit but does not fully meet PLOS ONE’s publication criteria as it currently stands. Therefore, we invite you to submit a revised version of the manuscript that addresses the points raised below during the review process.

We look forward to receiving your revised manuscript.

Kind regards,

Ray Borrow, Ph.D., FRCPath

Academic Editor

PLOS ONE

Reviewers' comments:

Reviewer's Responses to Questions

**Comments to the Author**

1. If the authors have adequately addressed your comments raised in a previous round of review and you feel that this manuscript is now acceptable for publication, you may indicate that here to bypass the “Comments to the Author” section, enter your conflict of interest statement in the “Confidential to Editor” section, and submit your "Accept" recommendation.

Reviewer #2: All comments have been addressed

Reviewer #3: (No Response)

2. Is the manuscript technically sound, and do the data support the conclusions?

Reviewer #2: Yes

Reviewer #3: Partly

3. Has the statistical analysis been performed appropriately and rigorously? 

Reviewer #2: Yes

Reviewer #3: I Don't Know

4. Have the authors made all data underlying the findings in their manuscript fully available?

Reviewer #2: Yes

Reviewer #3: No

5. Is the manuscript presented in an intelligible fashion and written in standard English?

Reviewer #2: Yes

Reviewer #3: Yes

6. Review Comments to the Author

Reviewer #2: Author has addressed all the comments that were raised by me during the previous two reviews and I am ok with the article.

Reviewer #3: The study by Yushuai Ma and colleagues suggest enhanced efficacy of immunization against Haemophilus influenzae by using of mannose-modified chitosan microsphere-based delivery system and outer membrane protein P6 antigen. mannose-modified chitosan or unmodified chitosan microspheres loaded with the recombinant P6 antigen was used to immunize mice and the immune response was evaluated. The authors report enhanced immune response elicited by the modified matrix delivery system, both systemic and mucosal response, in comparison with the unmodified version. The modified matrix assisted immunized mice show increased levels of IgG and IgA in serum and lavage fluid respectively. In addition, the modified matrix also enhanced the response of both humoral and cellular immunity response. Further, authors challenged the immunized mice with the bacterial suspension and observed the integrity of the lungs was maintained in the mice that received modified matrix P6 immunization, with lower infiltration of immune cells.

The authors argue that the observed immune response is because of the enhanced uptake of the microspheres by the macrophages through the mannose receptor mediated endocytosis. While the previous study (Zhu L, et al; ref# 43) suggests that this is a possibility, it is pivotal to show that P6-loaded chitosan upon modification behave as expected. The authors should check the invitro uptake of the microspheres by the receptor expressing cells and use a competition assay to show that it is specific to the receptors. All the immunological parameters show similar degree of differences between P6-CM and P6-MCM (except for the marginal difference in the antibody levels), suggesting that modification did not provide much benefit. So, it is essential to know that the modification is providing additional levels of protection to justify the study.

7. PLOS authors have the option to publish the peer review history of their article (what does this mean?). If published, this will include your full peer review and any attached files.

Reviewer #2: No

Reviewer #3: No

---

## [Author Response · Author response to Decision Letter 3]

4 May 2022

Reviewer #3:

1. The authors argue that the observed immune response is because of the enhanced uptake of the microspheres by the macrophages through the mannose receptor mediated endocytosis. While the previous study (Zhu L, et al; ref# 43) suggests that this is a possibility, it is pivotal to show that P6-loaded chitosan upon modification behave as expected. The authors should check the invitro uptake of the microspheres by the receptor expressing cells and use a competition assay to show that it is specific to the receptors. All the immunological parameters show similar degree of differences between P6-CM and P6-MCM (except for the marginal difference in the antibody levels), suggesting that modification did not provide much benefit. So, it is essential to know that the modification is providing additional levels of protection to justify the study.?

Answers:

In this experiment, mannose-modified chitosan microsphers are targeted to the mannose receptor of APC to enhance the humoral and cellular immune effects of the vaccine. The Ig /cytokines experimental datas and literature support (ref#19, 20, 43, 25-27) are sufficient to demonstrate that the mannose-modified chitosan microspheres can target MR in vivo although the direct experimental evidence cannot be supported. In detail: P6-MCMs and P6-CMs exhibited significant statistical differences except on humoral immunity IgA and IgG(Fig. 4). With the cellular immunity , P6-MCMs can trigger higher level of cytokines and CD4+/ CD8+ than P6-CMs(Fig. 5;6), The microsphere vaccines modified with mannose not only enhance cellular immunity but also trigger a mixed Th1/Th2-type immune response, while chitosan microsphere vaccines induce a Th2-type immune response. All these suggest that P6-MCMs enhance immune presentation by targeting APC mannose receptors, thereby enhancing immune responses. Most directly, the previous study (Jiang HL. ref#19 and Cui Z. ref#20) suggest that the mannose-modified chitosan microspheres really target the mannose receptor of RAW264.7 murine macrophage cells。

---

## [Editor Report · Decision Letter 4]

17 May 2022

Mucosal immunity of mannose-modified chitosan microspheres loaded with the nontypable Haemophilus influenzae outer membrane protein P6 in BALB/c mice

PONE-D-21-06673R4

Dear Dr. yutuo,

We’re pleased to inform you that your manuscript has been judged scientifically suitable for publication and will be formally accepted for publication once it meets all outstanding technical requirements.

Kind regards,

Ray Borrow, Ph.D., FRCPath

Academic Editor

PLOS ONE
---

## [Editor Report · Acceptance letter]

1 Jun 2022

PONE-D-21-06673R4 

Mucosal immunity of mannose-modified chitosan microspheres loaded with the nontyepable *Haemophilus influenzae* outer membrane protein P6 in BALB/c mice 

Dear Dr. Zhang:

I'm pleased to inform you that your manuscript has been deemed suitable for publication in PLOS ONE. Congratulations! Your manuscript is now with our production department. 

Kind regards, 

on behalf of

Prof. Ray Borrow 

Academic Editor

PLOS ONE